# Bipolar androgen therapy plus nivolumab for patients with metastatic castration-resistant prostate cancer: the COMBAT phase II trial

Cyclic high-dose testosterone administration, known as bipolar androgen therapy (BAT), is a treatment strategy for patients with metastatic castration-resistant prostate cancer (mCRPC). Here, we report the results of a multi-center, single arm Phase 2 study (NCT03554317) enrolling 45 patients with heavily pretreated mCRPC who received BAT (testosterone cypionate, 400 mg intramuscularly every 28 days) with the addition of nivolumab (480 mg intravenously every 28 days) following three cycles of BAT monotherapy. The primary endpoint of a confirmed $PSA_{50}$ response rate was met and estimated at 40% ($N = 18/45$, 95% CI: 25.7–55.7%, $P = 0.02$ one-sided against the 25% null hypothesis). Sixteen of the $PSA_{50}$ responses were achieved before the addition of nivolumab. Secondary endpoints included objective response rate (ORR), median PSA progression-free survival, radiographic progression-free survival (rPFS), overall survival (OS), and safety/tolerability. The ORR was 24% ($N = 10/42$). Three of the objective responses occurred following the addition of nivolumab. After a median follow-up of 17.9 months, the median rPFS was 5.6 (95% CI: 5.4–6.8) months, and median OS was 24.4 (95% CI: 17.6–31.1) months. BAT/nivolumab was well tolerated, resulting in only five (11%) drug related, grade-3 adverse events. In a predefined exploratory analysis, clinical response rates correlated with increased baseline levels of intratumoral PD-1 + T cells. In paired metastatic tumor biopsies, BAT induced pro-inflammatory gene expression changes that were restricted to patients achieving a clinical response. These data suggest that BAT may augment antitumor immune responses that are further potentiated by immune checkpoint blockade.

Androgen deprivation therapy (ADT) is the backbone of therapy for patients with metastatic prostate cancer. Despite extensive therapeutic targeting of the androgen receptor (AR), advanced prostate cancer commonly remains dependent on AR signaling[1]. One mechanism involves adaptive AR overexpression, whereby the AR protein is upregulated to compensate for a low testosterone environment[2].

Supraphysiologic levels of androgens may take advantage of increased AR expression and have utility in the treatment of metastatic castration resistant prostate cancer (mCRPC)[3–5].

In preclinical prostate cancer models, supraphysiologic testosterone inhibited cell cycle progression, induced double-strand DNA breaks and genomic rearrangements, and downregulated AR as well as

✉ e-mail: mmarko12@jhmi.edu

constitutively active AR splice variants[5–13]. Bipolar androgen therapy (BAT), whereby serum testosterone levels are rapidly driven to a supraphysiologic range followed by a return to near-castrate levels over 28-day cycles, is a treatment paradigm for patients with mCRPC developed by our group[3]. Multiple clinical trials have shown BAT to be well tolerated (not resulting in disease flares) with preliminary clinical activity[14–18]. The exact mechanism of action of BAT remains uncertain, and multiple mechanisms may be possible. For instance, double-stranded DNA breaks, genetic translocations and inhibition of DNA re-licensing can be induced by supraphysiologic androgens in prostate cancer models[6,11]. Patients with mCRPC that harbor underlying homologous recombination DNA repair gene mutations and/or pathogenic *TP53* mutations derive durable clinical benefit from BAT, further suggesting a role of BAT in inducing DNA damage[19–21]. In addition, BAT can downregulate c-MYC expression, which correlated with clinical outcomes in a recent study[22]. An enhanced, AR-regulated gene expression signature in pretreatment mCRPC tumor tissues also predicted favorable response to BAT, suggesting that there may be multiple BAT-mediated effects on mCRPC[22].

Immune checkpoint inhibition in prostate cancer has demonstrated limited clinical benefit to date[23–26]. Previously, we observed deep and durable clinical responses to immune checkpoint inhibition in several patients with microsatellite stable, low tumor mutational burden mCRPC, all of whom received prior BAT before obtaining an immune checkpoint inhibitor[27]. We hypothesized that BAT may prime mCRPC patients to respond favorably to subsequent immune checkpoint blockade. Here, we show the results of the prospective Phase 2 COMBAT study for patients with advanced mCRPC treated with BAT in sequence with nivolumab.

## Results

A total of 53 patients were screened for the COMBAT with 8 screen failures. Forty-five patients were enrolled and received at least one dose of investigational therapy. The clinical and pathologic characteristics of the cohort are shown (Table 1). The majority of patients had Gleason ≥9 disease (51.1%) and were treated with 2 or more lines of novel AR-targeted therapies (53.3%). Nearly half of the patients (44.4%) received prior taxane chemotherapy.

All patients received at least one dose of BAT. Eight patients received only BAT, and experienced disease progression prior to the start of nivolumab. These patients were included in the analysis. Of these 8 patients, the reasons for removal from study included: $N=1$ withdrew consent (noted to have unconfirmed PSA rise), $N=3$ PSA progression, and $N=4$ radiographic progression. Genomic and molecular testing on these patients is provided for reference (Supplementary Fig. 1A, B). The confirmed $PSA_{50}$ response rate was 40.0% ($N=18/45$, 95% CI: 25.7–55.7%, $P=0.02$ one-sided against the 25% null hypothesis) (Fig. 1A). Two additional patients achieved a 50% decline in PSA that was not confirmed by a subsequent measurement. One patient with a confirmed $PSA_{50}$ response had "progressive disease" as best objective response due to the presence a new metastatic lesion. This patient also had a 50% decrease in sum of target lesions coinciding with the $PSA_{50}$ response. 57.8% of patients achieved any degree of PSA decline from baseline. The $PSA_{50}$ response rate on BAT alone was $N=16/45=35.6\%$ (95% CI: 21.9–51.2%). In patients with measurable disease ($N=42$), the objective response rate was estimated at 23.8% (CR: $N=1$, PR: $N=9$, 95% CI: 12.1–39.5%) (Fig. 1B). Of note, $N=2$ patients who achieved a partial response had a PSA decline that did not reach criteria for a confirmed $PSA_{50}$ response. Best change in PSA and tumor volume for each patient is shown (Supplementary Fig. 2).

With respect to timing of PSA and/or objective responses, most responses were observed while on BAT monotherapy, with $N=2$ confirmed $PSA_{50}$ responses and $N=3$ objective responses having occurred following the addition of nivolumab. (Fig. 2). Ten patients had a decrease in PSA following nivolumab treatment (Supplementary Fig. 1A, C). We observed $N=4$ patients that had a decline in PSA on nivolumab after experiencing a PSA rise on BAT. $N=1$ of these 4 patients was noted to have an underlying mutation in a mismatch repair gene. Genomic and molecular data on these patients with a declining PSA on nivolumab are provided (Supplementary Fig. 1D). We did not observe an association between genomic alterations in key homologous recombination DNA repair genes or tumor suppressor genes and $PSA_{50}$ response rate (Supplementary Table 2).

The cut-off date for clinical data collection was 2/25/23 at which time all patients were off study. The median follow-up time was 17.9 months. The median rPFS was estimated at 5.6 (95% CI: 5.4–6.8) months (Fig. 3A). 11.1% ($N=5/45$) of patients remained on treatment for 11 months or longer. The median OS was 24.4 (95% CI: 17.6–31.1) months (Fig. 3B). We also report the median clinical or radiographic PFS (5.6 (95% CI: 4.8-6.0) months), median PSA PFS (4.0 (95% CI: 3.0–5.0) months), median duration of treatment (5.6 (95% CI: 5.6–8.4) months) (Supplementary Fig. 3). No significant difference in rPFS or OS was observed when stratified by prior chemotherapy (rPFS: No chemo – 5.8 months vs. chemo – 5.5 months; $P=0.55$; OS: no chemo – 28.2 months vs. chemo – 17.9 months; $P=0.82$) or number of lines of prior AR-targeted therapies (rPFS: 1 line of AR targeted therapy – 5.5 months vs. 2 or more lines – 5.8 months; $P=0.11$; OS: 1 line of AR targeted therapy – 24.4 months vs. 2 or more lines – 27.6 months; $P=0.6$) (Supplementary Table 3). Patients with visceral disease and

## Table 1 | Baseline Patient Demographic and Clinical Characteristics

|  | N = 45 |
|---|---|
| **Age (years)** | |
| Median | 69 |
| (Range) | (51–86) |
| **Race** | |
| Caucasian | 39 (86.7%) |
| African-American | 4 (8.9%) |
| Asian | 1 (2.2%) |
| Hispanic | 1 (2.2%) |
| **Baseline PSA (ng/mL)** | |
| Median | 57.6 |
| (Range) | (5.4-457) |
| **Gleason Sum** | |
| ≤ 7 | 11 (24.4%) |
| 8 | 8 (17.8%) |
| 9 | 18 (40.0%) |
| 10 | 5 (11.1%) |
| Unknown | 3 (6.7%) |
| **Visceral Disease** | |
| Yes | 7 (15.6%) |
| No | 38 (84.4%) |
| **Lines of Prior Novel AR Targeted Therapy** | |
| 1 | 21 (46.7%) |
| ≥2 | 24 (53.3%) |
| **Prior Taxane Chemotherapy** | |
| Yes | 20 (44.4%) |
| No | 25 (55.6%) |
| **≥ 2 Novel AR Targeted Therapy AND Prior Taxane Chemotherapy** | |
| Yes | 15 (33.3%) |
| No | 30 (66.7%) |

*N* Number of Patients, *AR* Androgen Receptor, *PSA* Prostate-specific antigen

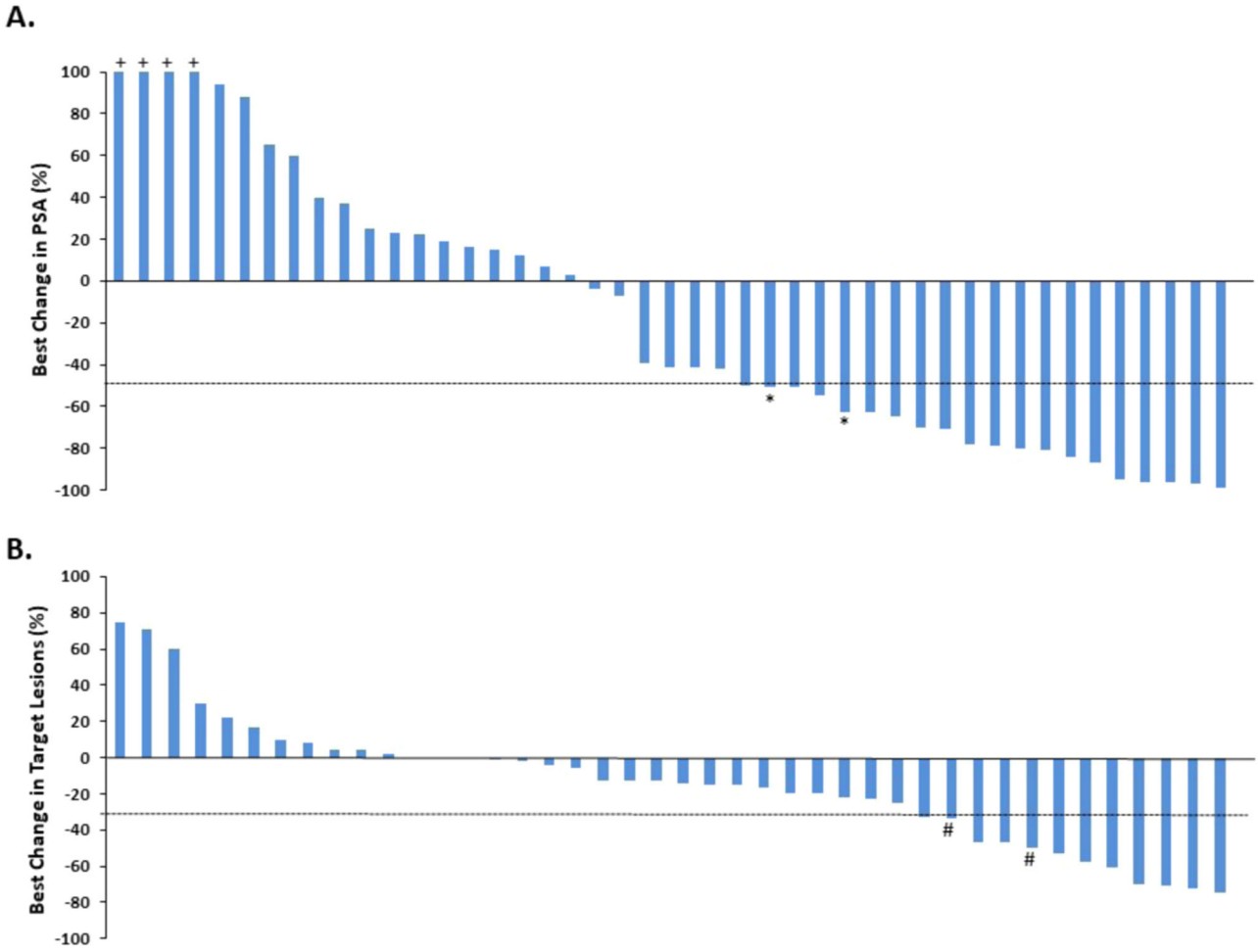

**Fig. 1 | PSA Response, and Objective Response: Best Change in PSA Level and Target Lesions in Patients with mCRPC Treated with BAT and Nivolumab.** **A** Waterfall plot showing the best PSA decline from baseline for each patient ($N = 45$). Eighteen ($N = 18$) patients were observed to have a confirmed PSA decline of 50% or greater while on treatment with BAT and nivolumab. Two additional patients achieved an unconfirmed $PSA_{50}$ response denoted by *. **B** Waterfall plot showing best change in tumor volume for $N = 42$ evaluable patients with measureable disease. Ten ($N = 10$) patients achieved a 30% or greater decrease in target lesions while receiving study treatment. # indicates progressive disease as best response due to new lesions. Source data are available as a Source Data file.

high Gleason sum at baseline had worse clinical outcomes (Supplementary Tables 4-5).

The majority of adverse events attributed to either BAT or nivolumab were Grade ≤2 (Table 2). Grade 3 musculoskeletal pain ($N = 1$; 2%) and lower extremity edema ($N = 1$; 2%) were attributable to BAT. Immune-related cardiac-specific Grade 3 events were observed in two patients (cardiomyopathy, pericarditis), which resulted in drug

### Table 2 | Treatment-Related Adverse Events Attributable to Bipolar Androgen Therapy or Nivolumab

|  | All Grades | Grade ≤ 2 | Grade 3 | Grade ≥ 4 |
|---|---|---|---|---|
| Musculoskeletal Pain | 15 (33%) | 14 (31%) | 1 (2%) | 0 |
| Nausea | 9 (20%) | 9 (20%) | 0 | 0 |
| Edema | 8 (18%) | 7 (16%) | 1 (2%) | 0 |
| Hypothyroidism | 5 (11%) | 5 (11%) | 0 | 0 |
| Rash | 5 (11%) | 5 (11%) | 0 | 0 |
| Lipase Elevation | 3 (7%) | 2 (4%) | 1 (2%) | 0 |
| Fatigue | 3 (7%) | 2 (4%) | 1 (2%) | 0 |
| Pericarditis | 1 (2%) | 0 | 1 (2%) | 0 |
| Cardiomyopathy | 1 (2%) | 0 | 1 (2%) | 0 |

Adverse events occurring in 10% or more of the patients and all Grade ≥ 3 events are shown. CTCAE version 4.0

discontinuation. One death occurred in the study from complications of COVID-19, and was not drug- or disease-related.

All patients underwent a mandatory soft tissue biopsy prior to the start of treatment (pre-treatment biopsy). A second biopsy was performed after 3 cycles of BAT, prior to the start of nivolumab (on-treatment biopsy), if this was deemed safe and feasible. We performed RNA sequencing (RNAseq) of laser-capture-microdissected, paired biopsies from $N = 12$ patients to interrogate BAT-mediated gene expression changes (additional paired biopsies were obtained, but some of the frozen tissue biopsies did not have any tumor, or did not have enough tumor for microdissection to generate adequate RNA for RNAseq). For these gene-expression analyses, patients were stratified by clinical response to BAT and nivolumab, defined as either a confirmed $PSA_{50}$ or an objective response. All $N = 12$ patients were treated with both BAT and nivolumab in the study. Accordingly, 6 patients were classified as responders and 6 patients were classified as non-responders. All 6 patients in the responder group achieved a $PSA_{50}$ and/or objective response prior to the addition of nivolumab. We next contrasted gene expression using gene set enrichment analysis (GSEA) of all HALLMARK gene sets between these two subgroups in biopsies obtained at baseline (Pre-BAT) or on C4D1 (On-BAT). BAT induced enrichment of gene sets associated with inflammation (i.e. Allograft Rejection, Interferon Gamma Response) in responders when comparing pre-treatment with on-treatment paired samples, but not in non-

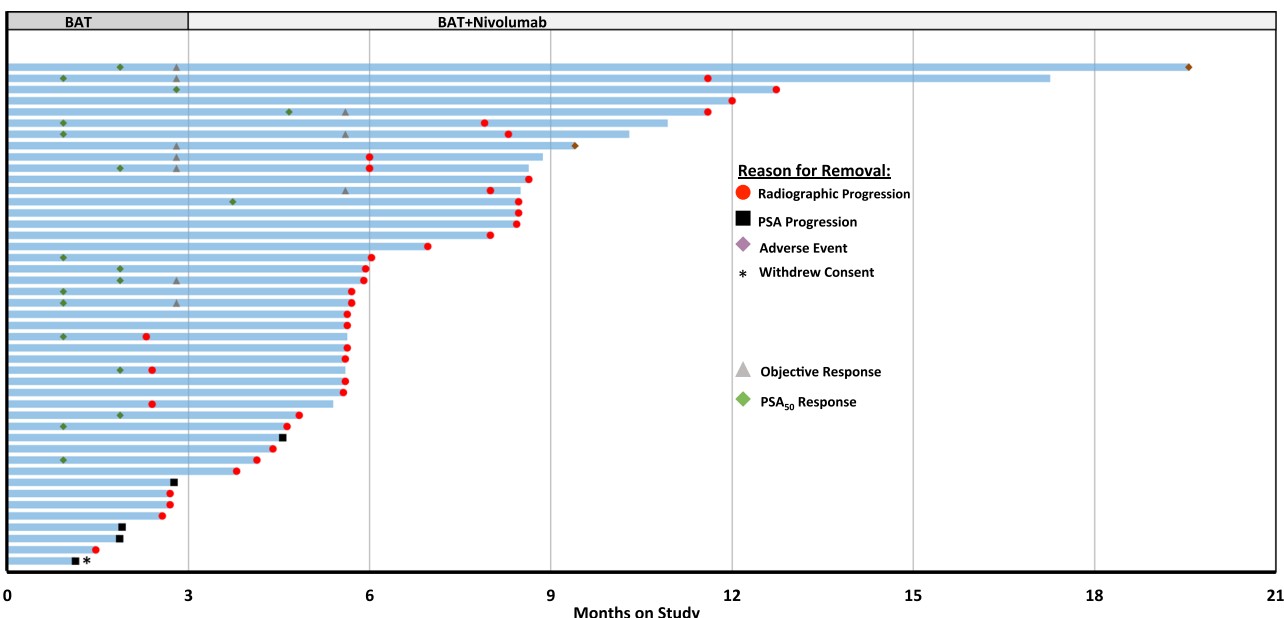

**Fig. 2 | Swimmers Plot of Patients with mCRPC treated on COMBAT study.** Duration of response demonstrated for each patient (*N* = 45) on COMBAT study. Patients were treated with BAT alone for the first 3 months on study. Nivolumab (continued with BAT) was initiated on Cycle 4 Day 1. Reason for study removal is listed. The timing of PSA and or objective responses are indicated. Source data are provided as a Source Data file.

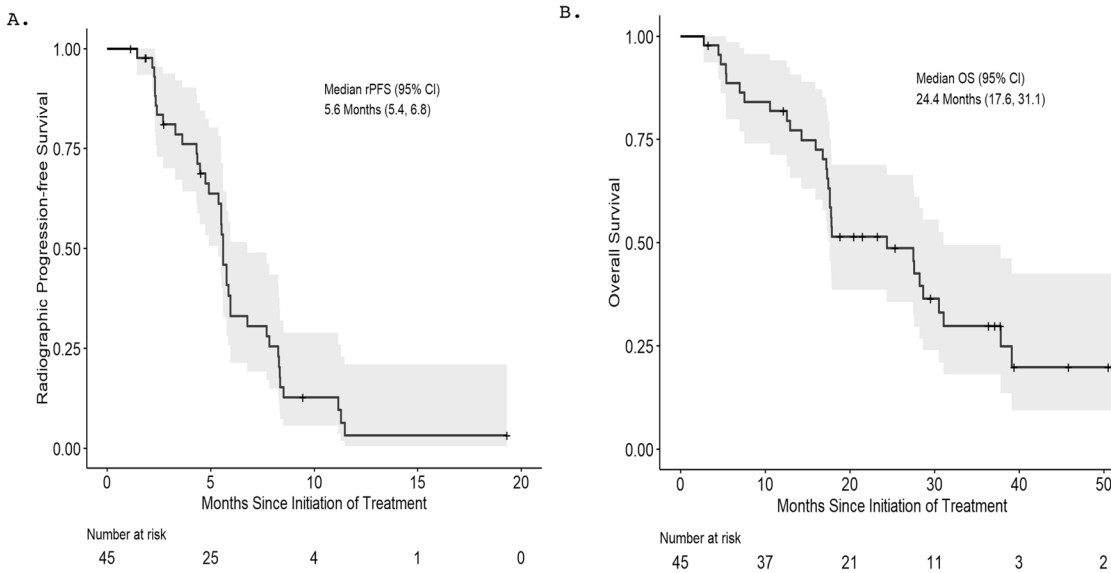

**Fig. 3 | Radiographic Progression-Free Survival and Overall Survival Estimates of Patients with mCRPC Treated with BAT in Combination with Nivolumab.** Kaplan-Meier curves of **A** radiographic progression-free survival (rPFS) and **B** overall survival (OS) are shown. The median rPFS was estimated at 5.6 months (95% Confidence interval: 5.4–6.8 months). The median OS was estimated at 24.4 months. (95% Confidence interval: 17.6 – 31.1 months). The shaded area represents 95% confidence region.

responders (Fig. 4A). The gene set that was most enriched by BAT in responding patients (pre-treatment vs. on-treatment) (Fig. 4B) and when comparing on-treatment samples between responders and non-responders (Fig. 4C) was ALLOGRAFT REJECTION. The top 30 transcripts based on rank metric scores that contributed to each enrichment are shown (Fig. 4B, C). GSEA using other gene sets are provided (Supplementary Fig. 4).

In a predefined analysis, we also performed quantitative, multiplex immunohistochemistry on paired FFPE biopsies to assess whether BAT alters the intratumoral density of T cells and/or whether intratumoral density of T cells predicts clinical response to BAT±nivolumab.

Twenty-three patients (*N* = 12 non-responders; *N* = 11 responders) with paired pre-treatment and on-treatment biopsies were evaluable for this analysis. Representative findings are shown in individual patients (Fig. 5A–L). In pretreatment biopsies, the median density of total CD8 + T cells trended higher in responders vs. non-responders (85.4 vs. 25.4 cells/mm², *P* = 0.07) (Fig. 5M). When probing further, the density of the PD-1+ subset of CD8+ cells was significantly higher in patients who achieved a response to BAT±nivolumab vs. those who did not (CD8 + PD-L1 + : 56.7 vs. 9.8 cells/mm², *P* = 0.03) (Fig. 5N). A similar effect was observed in overall CD4 + T cells (CD4 + : 158.5 vs. 66.0 cells/ mm², *P* = 0.10), as well as the CD4+ subset that expressed PD-1

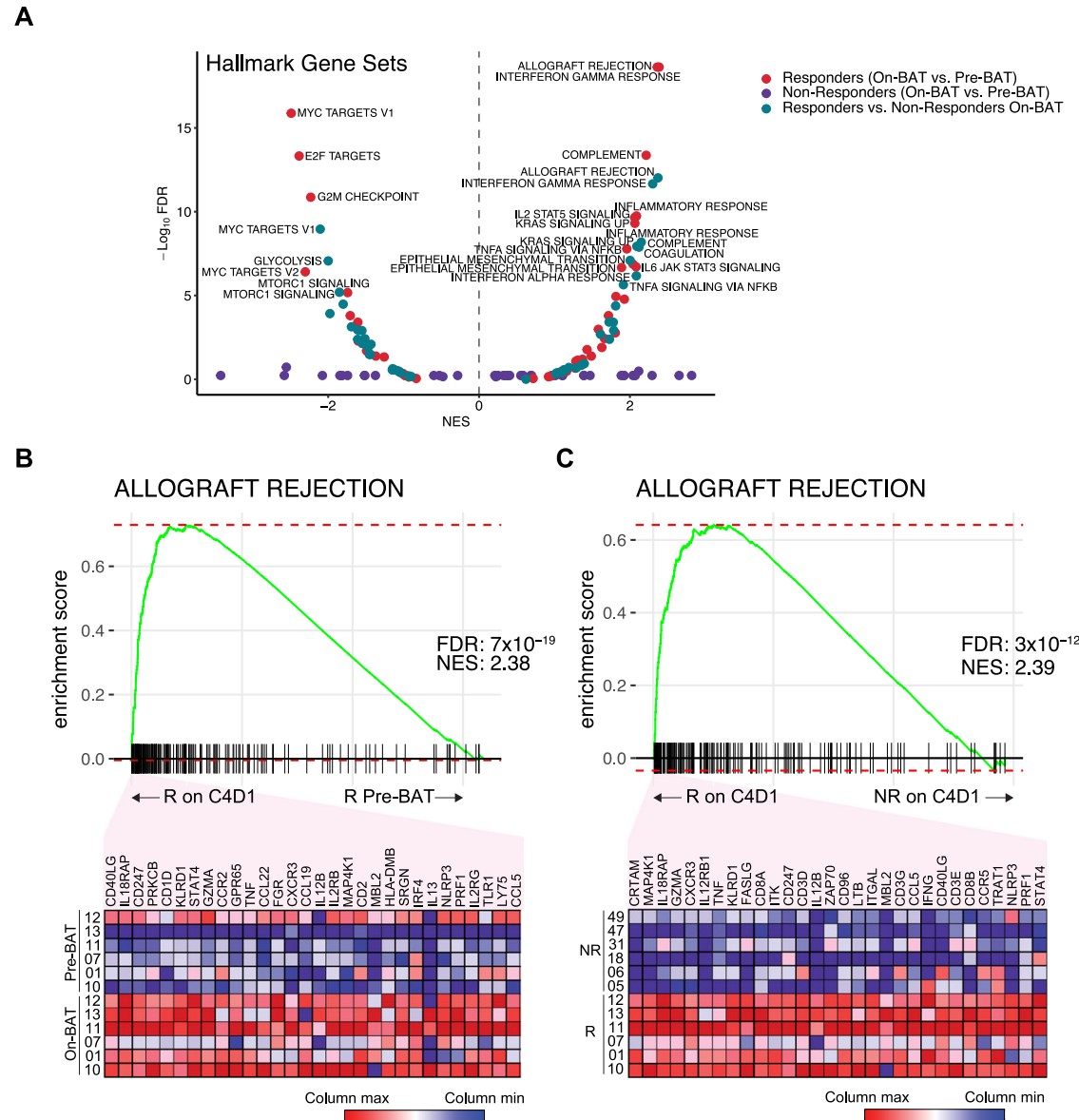

**Fig. 4 | BAT Induced Changes in Gene Sets Associated With Inflammation.**
**A** Gene set enrichment analyses of HALLMARK gene sets comparing baseline (pre-BAT) to C4D1 (On-BAT) in responders ($N = 6$; red dots) and non-responders ($N = 6$; purple dots) or responders to non-responders on C4D1 (On-BAT) (green dots). **B** Enrichment plot of ALLOGRAFT_REJECTION comparing pre-BAT to On-BAT in responding patients. Top 30 transcripts based on ranked metric scores are shown. **C** Enrichment plot of ALLOGRAFT_REJECTION comparing responders to non-responders on C4D1. Top 30 transcripts based on ranked metric scores are shown. FDR adj p, $p$-value adjusted for false discovery rate. ES, enrichment score. NES, normalized enrichment score. This is a two-tailed test where there is a multiple comparison adjustment for the statistics derived from the test (NES, FDR) based on the number of gene sets tested, number of samples and by applying numerous permutations of the gene list supplied and labels assigned to the samples.

(CD4 + PD-1+: 79.4 vs. 20.0 cells/mm², $P = 0.03$) (Fig. 5O, P). All PD-1+ T cells, as a group, were also higher in the responder cohort (Fig. 5Q). With respect to changes in T cell density upon BAT treatment, while some patient samples showed increases and others decreases, overall, there was no significant difference in the median density of CD8+ or CD4 + T cells irrespective of immune-cell PD-1 status (Supplementary Fig. 5).

## Discussion

In this Phase 2 study, we explored the combination of BAT given in sequence with nivolumab in heavily pretreated patients with mCRPC. We observed a confirmed PSA$_{50}$ response rate of 40%. This response rate compares favorably to findings in the TRANSFORMER study, a randomized Phase 2 study of BAT vs. enzalutamide in abiraterone-experienced patients with mCRPC, where the PSA$_{50}$ response rate to

BAT was 28%[18]. It should be noted that the 95% confidence interval of PSA$_{50}$ response rate in COMBAT does overlap with that in TRANS-FORMER. However, the COMBAT study included patients with more advanced diseases than the TRANSFORMER study (including those who had prior chemotherapy with no limit on prior AR-targeted therapies). This study also required soft tissue disease amenable to biopsy, which may not reflect the broader mCRPC population found in TRANSFORMER. Despite the addition of nivolumab, the median rPFS and median time on treatment (5.6 months) was similar to that observed in TRANSFORMER (5.7 months).

Since long-term outcomes on immune checkpoint inhibitors may not be reflected by PSA/objective response rates or rPFS[28,29], we also assessed overall survival. There is precedent in prostate cancer for overall survival benefit in the setting of low response rates and modest rPFS using immune based treatments. Sipuleucel –T did not show

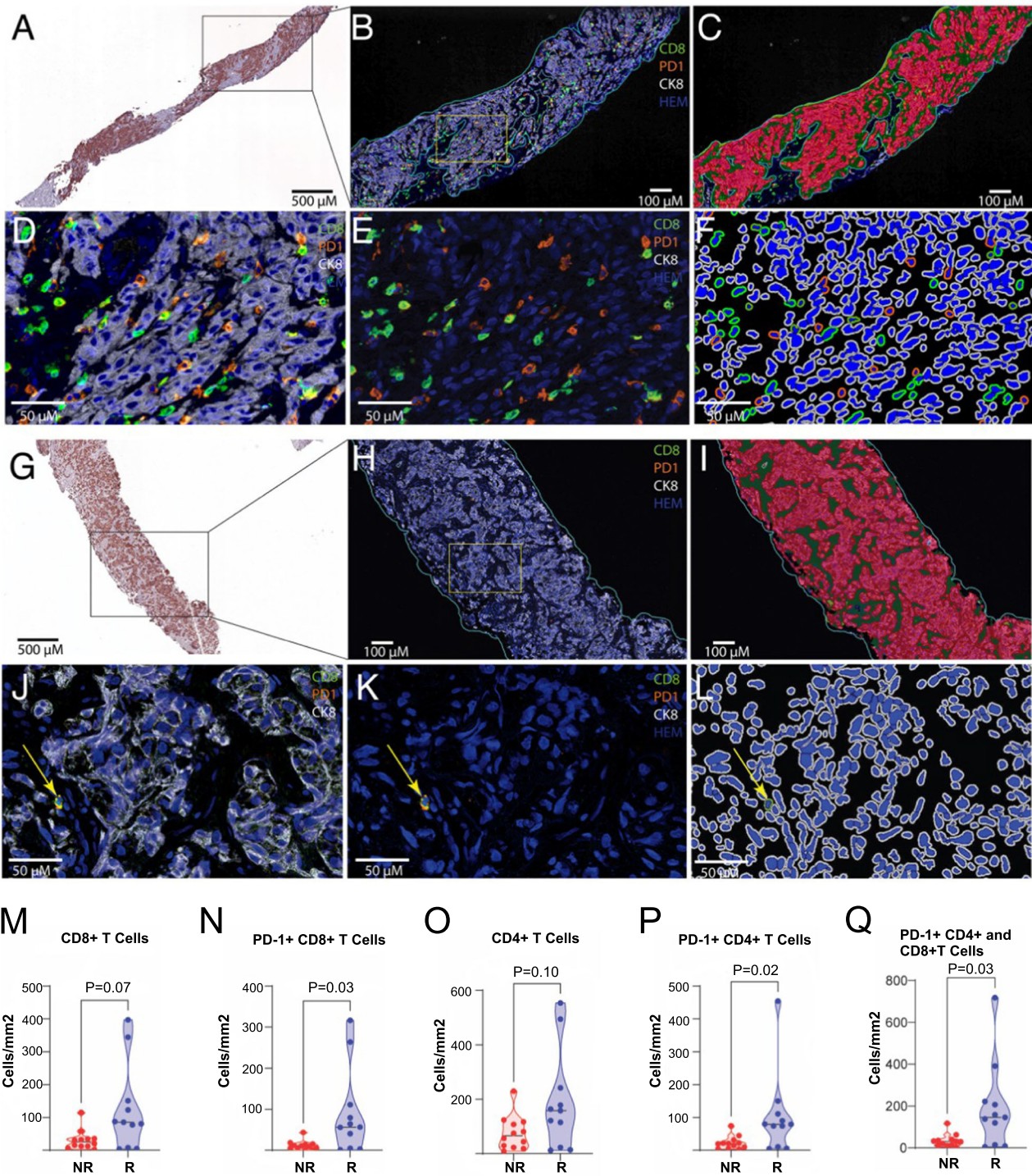

improvement of time to objective progression vs. placebo in patients with mCRPC[30]. However, overall survival benefit was observed in those patients receiving sipuleucel-T, which lead to its FDA approval. Despite a heavily-pretreated patient population, the estimated median OS in this study was greater than 2 years. These data included both chemotherapy naive and post-chemotherapy patients making broader extrapolation of these data difficult. Interestingly, we did not observe a difference in OS based on prior chemotherapy or number of prior AR-targeted therapies, suggesting that those with more advanced disease may still derive benefit from BAT±nivolumab. In patients who received at least one novel AR-targeted therapy and prior chemotherapy, median OS was estimated at 27.5 months. In the CARD study, patients previously treated with one AR targeted therapy and one taxane

chemotherapy has an estimated overall survival of 13.6 months after second line chemotherapy or 11.0 months following additional AR targeted therapy[31]. The FDA approval of Lu-PSMA and data from the VISION trial suggests these overall data from the CARD trial may be underestimated[32]. We caution that an OS analysis with 45 patients may be biased. However, a randomized prospective trial of BAT in combination with nivolumab would be required to validate any potential OS benefits in a post-chemotherapy setting.

We also sought to understand the pharmacodynamic effects of BAT on intratumoral molecular changes and the immune microenvironment as a means of explaining the favorable outcomes observed (i.e. PSA$_{50}$, OS). Although BAT did increase T cell density in a subset of patients (comparing matched on-treatment biopsies to

**Fig. 5 | Effect of BAT on T Cell Density in Metastatic Biopsies. A–F** shows images from a pretreatment biopsy from a responder showing relatively high levels of CD8-positive cells in the tumor. **A** Low power view of a pretreatment metastatic biopsy from stained for CK8. The boxed region is shown at higher power in **B, C**. **B** Pseudofluorescent image after color deconvolution and image fusions showing CD8 (green), PD1 (orange), CK8 (white; keratin 8 that stains all tumor cells) and hematoxylin (blue). The region of interest for T cell segmentation and quantification is circled by an outline in light blue to exclude non-tumor areas, including pre-existing lymph node (not shown here), and other tissues (e.g., liver when encountered). **C** This shows results after training of a random forest classifier, using CK8 as a guide, in which the epithelium is shown in red, the tumor stroma in green and yellow for empty space (non-tissue). **D** Shows higher power from the boxed area in **B** with CK8, CD8, PD1, and hematoxylin as in **B**. **E** This shows the same area as **D**, with the CK8 staining removed. **F, L** show cell segmentation for phenotyping. The negative cells show the nuclei in blue and the cell membranes in white. The CD8-positive cells (negative for PD1) are segmented with nuclei in blue and cell membranes in light green. PD1 positive/CD8 negative cells are segmented with nuclei in blue and cell membranes in orange. Double positive CD8 positive /PD1 positive cells are segmented with nuclei in blue and cell membrane in dark green. **G–L** Shows images from a pretreatment biopsy from a non-responder showing low levels of CD8-positive cells in the tumor. **G** Low power view of a pretreatment metastatic biopsy from stained for CK8. The boxed region is shown at higher power in **H, I. H** Pseudofluorescent image after color deconvolution and image fusions colored as above in **B. I** shows results after training of a random forest classifier as in **C. J–L** Higher power images of region shown in box in **H** and colored similar to D-F. Arrow shows a single CD8+ cell that is also positive for PD1. Median densities of T cell subtype stratified by responder (R)(N = 11) vs. non-responder (NR) (N = 12) are shown: **M** CD8 + T cells, **N** PD-1 + CD8 + T cells, **O** CD4 + T cells, **P** PD-1 + CD4 + T cells, **Q** PD-1 + CD8+ and CD4 + T cells. P values are derived from unpaired, two-sided t-tests without correction for multiple comparisons. The staining and quantitative approach is highly validated and performed once. Source data are provided as a Source Data file.

baseline biopsies), this finding did not correlate with clinical response rate. However, in those patients who achieved an objective or PSA response, baseline densities of PD-1 + CD4+ and PD-1 + CD8 + T cells were higher compared to patients that did not achieve a response. To try to explain the higher response rate with BAT +/- nivolumab observed here compared to BAT alone from historical studies, we speculated that BAT may induce a gene expression signature, which may synergize with this higher density of PD-1 + T cells. Consistent with this hypothesis, in patients achieving a clinical response to sequential therapy, BAT upregulated a pro-inflammatory gene signature, which has been associated with favorable response to immune checkpoint inhibition in other malignancies[33]. This suggests that BAT may take advantage of existing intratumoral T cell populations through direct tumoral molecular changes to generate clinical benefit. It is also possible that the addition of nivolumab may activate these PD-1 + T cells further enhancing an anti-tumor effect. In previous studies, we also observed BAT induced chemokines and cytokines secreted by tumor cells resulting in increased migration of immune cells to the tumor microenvironment[34]. This study provides further evidence that intratumoral, pro-inflammatory changes induced by BAT yields better clinical outcomes.

It has been also suggested that AR inhibition may improve immune checkpoint blockade effectiveness in models of prostate cancer via increased interferon-gamma expression in CD8 + T cells[35]. However, here we did not observe a decrease in *IFNG* abundance in tumors following BAT. In fact, we observed an enrichment of the HALLMARK INTERFERON GAMMA RESPONSE gene set in responding patients following BAT. This suggests that androgen exposure in BAT does not completely disable the ability of T cells to produce interferon-gamma. A limitation of our study is that we performed bulk, rather than single-cell, RNA sequencing so we could not evaluate a more complete T cell-specific transcriptional signature. Although androgens may affect T cell function[36], we speculate that the rapid cycling of testosterone levels, as seen with BAT, may result in a different phenotypic outcome with respect to T cell function. It is possible that stable levels of physiologic androgens may produce a more antagonistic effect on immune checkpoint inhibitor efficacy. A randomized Phase III trial of enzalutamide and pembrolizumab in patients with mCRPC (KEYNOTE-641) was discontinued due to futility after an interim analysis. Despite preclinical data suggesting a potential relationship between AR antagonism and response to immune checkpoint blockade, a prospective clinical trial did not support these findings.

In conclusion, the COMBAT study demonstrated that BAT in combination with nivolumab is safe and well tolerated in patients with advanced mCRPC. The high PSA$_{50}$ response rate, objective response rate, and durable overall survival observed suggest that further study in a randomized clinical trial is warranted. We propose testing our findings in a three arm, randomized Phase II trial including BAT alone, BAT in combination with nivolumab, and best standard of care. We also provide preliminary evidence to suggest a possible priming of the antitumor immune response by BAT, which may be exploited by subsequent administration of a checkpoint inhibitor.

## Methods
### Study cohort, design, and outcome measures
COMBAT is an Institutional Review Board (IRB)-approved, single-arm, multi-center Phase 2 trial that was conducted at Johns Hopkins Hospital in Baltimore, MD, USA and two other sites (Dana-Farber Cancer Institute in Boston, MA, USA and the University of California-San Francisco Hellen Diller Comprehensive Cancer Center in San Francisco, CA, USA). This trial was registered with ClinicalTrials.gov on 6/13/18 (https://classic.clinicaltrials.gov/ct2/show/NCT03554317). The study design and conduct complied with all relevant regulation regarding the use of human study participants and was conducted in accordance with the criteria set by the Declaration of Helsinki. Informed consent was obtained from all participants. Eligible patients were 18 years or older with mCRPC defined per Prostate Cancer Working Group 3 (PCWG3) guidelines[37]. All patients must have been treated with at least one novel AR-targeted therapy (i.e. abiraterone acetate, enzalutamide, or equivalent). Up to one taxane chemotherapy for the treatment of mCRPC was permitted. Serum testosterone levels ≤50 ng/dL and adequate bone marrow, renal, and liver function were required. All patients were required to have soft tissue disease amenable to a prospective metastatic tumor biopsy at baseline. A second paired biopsy was required after 12 weeks of BAT (i.e. at C4D1), if this was safe and feasible. At each biopsy, we attempted to obtain two tissue cores for formalin fixation and paraffin embedding (FFPE) and two that were snap-frozen. Patients were excluded if they had ≥5 sites of visceral disease, were on opioid analgesics for tumor-related pain, or were at risk for urinary obstruction or spinal cord compression as determined by the treating physician. Between September 5th, 2018 and October 27th, 2020, N = 45 patients were enrolled in the study.

All patients received intramuscular injections of 400 mg testosterone cypionate on day 1 of 28-day cycles. Patients were maintained on a luteinizing hormone-release hormone agonist/antagonist for the duration of the study in order to suppress endogenous testosterone. After 12 weeks on testosterone cypionate (3 doses of BAT), patients were treated with the addition of nivolumab 480 mg intravenously every 28 days and were maintained on concurrent testosterone cypionate. Patients were treated until clinical or radiographic progression, unless removed from study due to treatment-related toxicity.

The primary endpoint for this study was the PSA$_{50}$ response (i.e. ≥50% decline in PSA from baseline, and confirmed with a second measurement at least 4 weeks later) at any time on BAT±nivolumab.

Secondary endpoints included objective response rate (ORR), radiographic progression-free survival (rPFS), overall survival (OS), PSA progression-free survival (PSA PFS), clinical or radiographic progression-free survival (crPFS), median duration of treatment and safety of BAT±nivolumab.

The ORR was defined as the percentage of patients who achieve an objective response (complete response or partial response) by RECIST 1.1 criteria among those with measurable disease. rPFS was the time from the date of first dose to the date of radiographic progression per RECIST 1.1 criteria for soft tissue lesions and PCWG3 guidelines for bone lesions, or death, whichever occurred first. Patients who were alive and whose disease did not progress at the end of follow-up were censored at the date of the last tumor assessment. PSA PFS was defined as the time from the date of the first dose to the time of PSA progression according to PCWG3 criteria (increase from PSA nadir $\geq 25\%$ and by $\geq 2$ ng/mL). crPFS was defined as the time from the date of the first dose to clinical progression or radiographic progression according to RECIST 1.1 for soft tissue disease or PCWG3 for bone lesions. OS was defined as the time from the first dose to death from any cause. Patients without death event were censored at the last known alive date. Toxicities were assessed according to CTCAE version 5.

## Statistical analysis
The null hypothesis for the primary endpoint was a $PSA_{50}$ response rate of 25% based on historical clinical trials using BAT. The alternative hypothesis was a $PSA_{50}$ response rate of 45% to BAT±nivolumab. A Simon's two-stage minimax design was planned and a sample size of 39 patients had 90% power to reject the null $PSA_{50}$ response rate of 25% in favor of 45% response rate with a one-sided type I error of 0.1. Allowing for 10% possible dropout rate and unevaluable patients, enrollment of 44 patients was planned, and 45 were enrolled because the last two patients consented at two sites simultaneously. $PSA_{50}$ response rate and ORR were estimated as proportions, and the corresponding exact 95% confidence intervals were estimated using Clopper–Pearson method. Kaplan-Meier method was used to summarize rPFS and OS. The associations of each patient baseline characteristic with rPFS and OS were evaluated using univariate Cox regression models. The difference in baseline levels of T cell densities between responders and non-responders were compared using Wilcoxon rank sum test. The change in T cell densities after BAT from baseline was assessed using Wilcoxon signed rank test. All statistical analyses were performed using R software (Version 4.2.1). All tests were two-sided unless otherwise noted, and statistical significance was set at $P < 0.05$, except for the primary endpoint, as per design, statistical significance was one-sided $P < 0.1$."

## RNA sequencing
For RNA sequencing of patient biopsy samples, regions of cancer were laser capture microdissected from fresh frozen tissue biopsies and purified RNA was provided to the SKCCC Experimental and Computational Genomics Core to carry out low-input RNA-seq workflow as described previously with some modifications[38]. Briefly, the quality of total RNA was measured by the Agilent Bioanalyzer to determine RNA integrity (RIN). Samples with starting input between 100pg and 100ng of total RNA and RIN > 7.0 were considered to have sufficient quality to proceed to construction of whole transcriptome sample-barcoded libraries using the Ovation RNA-Seq System V2 according to the manufacturer's protocols (Nugen). Quantification of the libraries was performed by qPCR or by the Agilent Bioanalyzer and equimolar concentrations of each library were pooled together, clustered and sequenced on an Illumina Novaseq 6000 platform, with paired end sequencing. The resulting reads were aligned to the human reference genome build hg38 using STAR aligner[39] and quantified with RSEM[40] to obtain read count estimates for each gene, which were then normalized and log2 transformed using DESeq2[41].

## Gene set enrichment analysis (GSEA)
Differentially expressed genes (DEG) were estimated using DESeq2 based on matched biopsy pairs comparing patient groups stratified according to treatment response and time point of biopsy collection. Log2 Fold changes in gene expression were then used to rank DEG and GSEA performed using fGSEA[42] in R. Hallmark gene sets and ImmuneSigDB were downloaded from the Molecular Signatures Database (MSigDB), with the latter restricted to sets associated with exhaustion. Nanostring gene sets were obtained from the 770 nCounter® Pan-Cancer Immune Profiling Panel from Nanostring, (https://nanostring.com/products/ncounter-assays-panels/oncology/pancancer-immune-profiling/) and grouped into the categories described by the manufacturer. Enrichment scores and p values corrected for false discovery are reported within the figures.

## Multiplex IHC and image analysis
Iterative chromogenic multiplex IHC was performed for CD3, CD4, CD8, PD1, FOXP3, and CK8 as described[43] on 24 sets of matched pretreatment and C4D1 treatment biopsies that had adequate tissue for staining. Whole slide scanning, image registration, color deconvolution, image fusion, and phenotypic image analysis was performed as previously described[43]. Antibodies with conditions: PD1 Source: Abcam Species: Mouse Monoclonal Clone: NAT105 Dilution: 1:200 Incubation: ON* 4 °C Antigen retrieval: TR*** Secondary Kit: UltraVision Quanto (Leica). CD3 Source: DAKO Species: Rabbit Polyclonal Dilution: 1:600 Incubation: 45 min RT** Antigen retrieval: Citrate Secondary Kit: PowerVision+ (Leica PV6119). CD8 Source: DAKO Species: Mouse Monoclonal Clone: C8/144B Dilution: 1:200 Incubation: 45 min RT Antigen Retrieval: Citrate Secondary kit: PowerVision+ (Leica PV6114). CD4 Source: Abcam Species: Rabbit Monoclonal Clone: EPR6855 Dilution: 1:2000 Incubation: 45 min RT Antigen Retrieval: Citrate Secondary Kit: PowerVision+ (Leica PV6119). FOXP3 Source: eBioscience Species: Mouse Monoclonal Clone: 236 A/E7 Dilution: 1:250 Incubation: ON 4 °C Antigen retrieval: TR Secondary Kit: PowerVision+ (Leica PV6114).

For quantifications of T cell types, to eliminate potential immune cells that were already present in the metastatic sites, we manually annotated tumor region of interest, using CK8 staining as a guide, and included a small border (approximate 250 μm). For each cell phenotype (Supplementary Table 1; $N = 8$), we quantified the density of cells with that phenotype per unit area in the regions of interest as previously described[43].

## Whole genome sequencing of DNA from laser capture micro-dissected cancer regions from biopsy tissues
Fresh frozen blocks of baseline biopsy tissues were subjected to laser capture micro-dissection (LCM) for enrichment of cancer regions, and subjected to DNA isolation as we described in detail previously (Sena et al., JCI, 2022)[22]. Briefly, frozen sections mounted on polyethylene naphthalate (PEN) slides (Leica) were stained with hematoxylin and then subjected to LCM enrichment of regions highly enriched for tumor cells at the SKCCC Cell Imaging Core using a Leica LMC7000 system. DNA and RNA was isolated using the ALLPREP RNA/DNA extraction method (Qiagen). DNA amount was quantified using the Qubit dsDNA HS assay (Thermo Fisher). DNA isolated from matched whole blood was used as matched normal. DNA from Tumor and normal pairs were then subjected to barcoded whole genome sequencing library preparation using the TruSeq DNA Nano Library Prep kit (Illumina), and resulting libraries were sequenced using an S4 flow cel on a NovaSeq 6000 instrument (Illumina), with paired-end 150 bp×2 chemistry, at a raw per sample throughput of approximately 110 gigabases (producing a raw average sequencing depth of 30× haploid genome coverage). The resulting reads in fastq files were trimmed using Trimgalore (v0.6.7) software to trim off adaptor sequences and low-quality bases. The optimized Sentieon pipeline

(release 202010.02) was used to: (i) implement alignment of reads to the hg38 reference genome using bwa mem (v0.7.17), and then aligned bam files were further processed to create a recalibrated bam file (using the following tools: Piccard-tools MarkDuplicates v2.9.0, GATK IndelRealigner v3.8.0, GATK BaseRecalibrator v3.8.0, GATK PrintReads v3.8.0); (ii) identify germline variants using GATK HaplotypeCaller (v3.8.0) on files from the matched normal tissues; (iii) identify somatic variants using GATK MuTect2 (v3.8.0) between tumor-normal pairs. The resulting variant VCF files were annotated and converted to mutation annotation format using vcf2maf (v1.6.19) using variant effect predictor (vep) annotation. Somatic structural variants were identified using Manta (v1.6.0), and somatic copy number alterations were identified using CNVkit (v0.9.4). Somatic alterations affecting a set of known recurrently mutated prostate cancer driver genes identified in prior studies were filtered and reported. [For copy number alterations, only genomic segments/genes with deep deletions and high gain (abs(log2ratio) > 1.0) were considered significant. For known tumor suppressor genes, loss of function truncating mutations and mutations annotated in OncoKB as pathogenic/likely pathogenic were considered significant. For known oncogenes, alterations identified as pathogenic/likely pathogenic by OncoKB were considered significant.]

### Reporting summary

Further information on research design is available in the Nature Portfolio Reporting Summary linked to this article.

## Data availability

The minimum datasets necessary to interpret this research have been provided within the Article, Supplementary Information and Source Data File, where applicable. The raw RNAseq data are protected and are not available due to data privacy laws. The processed RNAseq data are available at the NCBI's Gene Expression Omnibus (GEO) under the accession code GSE229555. We did not consent patients for the public release of raw WGS data. We used the somatic mutation calls derived from WGS in this analysis. We have provided the source data listing these somatic mutation calls in the Source Data file. The study protocol is available in the Supplementary Information file. Additional datasets generated and/or analyzed during the current study, including de-identified participant data, can be made available from the corresponding author on request. Source data are provided in this paper.

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

## Acknowledgements

The project described was supported by the Sidney Kimmel Comprehensive Cancer Center at Johns Hopkins NIH grant P30 CA006973, R01 CA184012, R01CA243184, Department of Defense W81XWH-19-1-0692 and W81XWH-19-1-0724, and a Prostate Cancer Foundation Challenge Award.

The content is solely the responsibility of the authors and does not necessarily represent the official views of the National Cancer Institute or the National Institutes of Health.

## Author contributions

Study conception and design: E.S.A., M.C.M., S.R.D., Project. supervision: E.S.A., S.R.D., A.M.D., Participant recruitment and coordination: M.C.M., M.E.T., R.A., V.S., L.A.S., A.L., C.J.P., M.A.C., C.H.M., M.A.E., E.S.A., S.R.D. Data collection, processing, and analysis: M.C.M., M.E.T., R.A., L.A.S., H.Q., H.W., B.O., T.J., C.G.-A., S.Y., D.E.S., A.M.D., E.S.A., S.R.D. Manuscript preparation: M.C.M., E.S.A. with all authors contributing to manuscript review and editing.

## Competing interests

M.C.M. is a paid consultant to Clovis Oncology and Exelixis. E.S.A. has served as a paid consultant for Janssen, Astellas, Sanofi, Bayer, Bristol Myers Squibb, Amgen, Constellation, Blue Earth, Exact Sciences, Invitae, Curium, Pfizer, Merck, AstraZeneca, Clovis, and Eli Lilly; has received research support (to his institution) from Janssen, Johnson & Johnson, Sanofi, Bristol Myers Squibb, Pfizer, AstraZeneca, Novartis, Curium, Constellation, Celgene, Merck, Bayer, Clovis and Orion; and is a co-inventor of a biomarker technology that has been licensed to Qiagen. R.R.A. has served as a paid consultant for Janssen, Bayer, Pfizer, Amgen, Merck, AstraZeneca, Lumanity, Deallus, OncLive, EcoR1, Novartis; has received research support (to his institution) from Janssen, Merck, AstraZeneca, Zenith Epigenetics, Amgen, Novartis. The remaining authors declare no other competing interests.

## Additional information

**Mark C. Markowski** [1] ✉, **Mary-Ellen Taplin**[2], **Rahul Aggarwal** [3], **Laura A. Sena**[1], **Hao Wang** [4], **Hanfei Qi** [4], **Aliya Lalji**[1], **Victoria Sinibaldi**[1], **Michael A. Carducci**[1], **Channing J. Paller**[1], **Catherine H. Marshall**[1], **Mario A. Eisenberger**[1], **David E. Sanin** [1], **Srinivasan Yegnasubramanian** [1,5], **Carolina Gomes-Alexandre**[5], **Busra Ozbek**[5], **Tracy Jones**[5], **Angelo M. De Marzo** [1,5], **Samuel R. Denmeade**[1] & **Emmanuel S. Antonarakis**[1,6]

[1]Department of Oncology, Sidney Kimmel Comprehensive Cancer Center, Johns Hopkins University, Baltimore, MD, USA. [2]Department of Medical Oncology, Dana-Farber Cancer Institute, Boston, MA, USA. [3]Helen Diller Family Comprehensive Cancer Center, University of California San Francisco, San Francisco, CA, USA. [4]Division of Quantitative Sciences, Department of Oncology, Johns Hopkins School of Medicine, Baltimore, MD, USA. [5]Department of Pathology, Johns Hopkins School of Medicine, Baltimore, MD, USA. [6]Department of Medicine, Masonic Cancer Center, University of Minnesota Medical Center, Minneapolis, MN, USA. ✉e-mail: mmarko12@jhmi.edu

