## [Peer Review File · Nature Communications]

Bipolar Androgen Therapy Plus Nivolumab for Patients with Metastatic Castration-Resistant Prostate Cancer: the COMBAT Phase II TrialEditorial Note: This manuscript has been previously reviewed at another journal that is not operating a transparent peer review scheme. This document only contains reviewer comments and rebuttal letters for versions considered at *Nature Communications*. Mentions of prior referee reports have been redacted.

REVIEWER COMMENTS

Reviewer #1 (Remarks to the Author):

Major comments

1. The authors should make clear in the abstract the response rate to BAT alone and BAT with nivolumab.
2. The authors should not claim the combination of BAT and nivolumab to be 'safe'; this trial's size cannot define this. A better term is 'well tolerated'.
3. The authors should be clear in the abstract the second biopsy was taken after BAT alone and before nivolumab.
4. The authors should also make clear in the abstract there was a lack of correlation between BAT response and DNA repair defects, which they previously suggested.
5. The authors report that BAT induced changes in inflammatory pathways with the top pathway being the IL6-JAK-STAT3 pathway. It has been reported that IL6-JAK-STAT3 also has immunosuppressive functions on T cell and possibly drive cancer progression and treatment resistance. They should comment on this in their discussion. It would be important to show how BAT treatment impacted: 1) T cell exhaustion signatures in the RNAseq data; 2) Other inferred immune cell types; 3) Broader immune transcript expression using 770 gene Nanostring White paper.
6. Authors report on the BAT induced inflammatory response; can they please comment on impact on T cell density (Figure 5). What does this mean for T cell anti-tumour function and the impact of BAT on nivolumab treatment outcome?
7. One of 4 responders to nivolumab appears to have a mutation in an MMR gene. Did mutational burden/MMR mutational signature result in responders?

Minor comment

Typo on page 62 of the PDF: Cell Phenotypes "Evalutated" in T cell Panel

4. Does PDL1 express in tumour, especially in responders please?

[**Editorial Note:** prior referee report redacted]

Reviewer #5 (Remarks to the Author):

1) I think it is difficult to assess if the sequence BAT plus nivolumab has some clinical benefit as measured by response, given that most responses were observed in the BAT period alone. I think the trial should aim to show whether adding nivolumab extends time to progression or make those responses more durable, so maybe a randomised phase II of BAT vs BAT+Nivo trial and PFS endpoint would have reflected this more clearly. In these lines, maybe you want to add description of duration of responses (it is, as a minimum, defined for RECIST 1.1 in the guidelines, and you could also define for PSA). The swimmers plot helps hugely to understand the timing and duration of responses. It is also worth noting that 5 patients continue on treatment beyond radiological progression for a few weeks more – this is not uncommon in this disease (and particularly if patients are heavily pre-treated and don't have many options after). This is also highlighted in the PCWG3 guidelines, where the metric "no longer clinically benefiting (NLCB)" was introduced, defined as the date and the specific reason(s) a therapy was ultimately discontinued. It is somewhat subjective, but I think it would be worth reporting median time on treatment, as it may be more informative as a measure of clinical benefit. Please also add some data re compliance (dose reductions, whether patient could continue on one treatment only, did it happen?).

2) The trial has a 2-stage Simon design, but some details are missing (e.g. stage 1 and stage 2 success criteria, specifying this is a Minimax design). The statistical methodology used for the primary endpoint (95% exact confidence intervals used and p-value as a one-sample binomial test) does not take into account the 2-stage design – please note the following reference: Koyama T and Chen H (2008): Proper inference from Simon's two-stage designs. *Statistics in Medicine*, 27(16):3145-3154.

Appropriate p-value and confidence intervals could be calculated with the OneArmPhaseTwoStudy R package or clinfun R-package. For your trial, I have done these calculations using these packages, and it won't make a huge difference (p-value of 0.018),

because the number of responses observed is well above the threshold of success for the design. But I wanted to make you aware as, if you had observed number of responses around the threshold of success, you could see some inconsistencies between the one-sample binomial test p-value and the two-stage p-value (like one-sample binomial p-value not significant when min number of responses was achieved). A similar issue occurs for the confidence interval. These calculations also take into account if over-recruitment occurs, as it has been the case in the COMBAT trial.

3) Did some of these patients present with bone disease (in addition to soft tissue)? Should this be indicated in the baseline table? rPFS is defined as time to PD as per RECIST 1.1 in the manuscript, but the PCWG3 guidelines include progression on bone scan also as radiographic PD. Could you clarify if this is the case? Of note, the protocol specifies also as an endpoint PFS, a combination of clinical and radiographic progression (including progression on bone scan), which is not reported.

4) For these gene-expression analyses of the paired tissue samples, patients were stratified by clinical response to BAT and nivolumab (as either PSA50 or OR by RECIST). But the second biopsy is before nivolumab, so it would be worth clarifying/stating if these 6/12 responders did respond while on BAT only, or after when already on BAT+nivo, and maybe discuss results taking this into account. Would it be more appropriate to say then that “patients were stratified by their response to BAT”, only?

5) In the comparison of COMBAT PSA response rates with TRANSFORMER, it may be helpful to report the PSA50 rate while on BAT only (16/45=35.6%), with its corresponding confidence intervals, which would more clearly overlap with those in the TRANSFORMER study. As discussed above, in addition to rPFS may be worth to report median time on treatment and compare this to the TRANSFORMER study.

Minor:

- Maybe plotting per-patient PSA dynamics (line or spaghetti plots) may also help visualise differences of PSA kinetics while in BAT only or while in the combination. Re Figure 1, it may help to have the same patients aligned, so the reader can map out per-patient max tumour

shrinkage and max PSA decrease. The swimmer plot summarises nicely the time of the first decrease to qualify as a response, but not the magnitude of it. You could chose the order of the patients in the current PSA waterfall plot for the tumour shrinkage plot.

- Methods line 211 “The associations of patient baseline characteristics with rPFS and OS were evaluated using univariate Cox regression model.” could be changed to “The associations of EACH patient baseline characteristic with rPFS and OS were evaluated using univariate Cox regression models”, so it is clear these are separate Cox models for each factor.

- Methods line 216: “All tests were two-sided unless otherwise noted, and statistical significance was set at $P < 0.05$ ” – this is except for the primary endpoint, as per design, statistical significance is one-sided $P < 0.1$.

- Results line 306: it says 8 patients experienced PD prior to start of nivolumab, but in the reasons for removal from study, for one patient reason is “Withdrew consent”. Assume it is withdrew consent AND progression? Was it a clinical progression without PSA and radiological progression?

- Results line 314: “One patient with a confirmed PSA50 response had “progressive disease” as best objective response due to the presence a new metastatic lesion.” Worth explaining timing of PSA respect timing of radiological progression, where these observed at the same time? Were target/non-target lesions responding/stable?

- Results line 318: a CR was observed, did patient had bone disease at baseline?

- Results line 321: I think it would be best to state upfront that, “Most responses were observed while on BAT monotherapy, with only N=2 confirmed PSA50 responses and N=3 objective responses occurred following the addition of nivolumab. (Figure 2)”.

- Results line 322: Consider adding “further” - “Ten patients had a FURTHER decrease in PSA following nivolumab treatment (Extended Data Figure 1A, C).”

- Results line 324: “We observed N=4 patients that had a decline in PSA on nivolumab after experiencing a PSA rise on BAT”. In Extended Data Figure 1, I only count 2 such patients (20&33), is this correct?

- Results line 333: “11.1% (N=5/45) of patients remained on study for 12 months or longer.” What do you mean, “on study”? “On study treatment”? I imagine patients are still on study after treatment discontinuation for OS. Also, note I count 4 patients on treatment for 12 months or longer in the swimmers plot, not 5.

A point-by-point response to each reviewer's comments is listed below.

Reviewer #1 (Remarks to the Author):

Major comments

1. The authors should make clear in the abstract the response rate to BAT alone and BAT with nivolumab.

RESPONSE: We have made this addition.

2. The authors should not claim the combination of BAT and nivolumab to be 'safe'; this trial's size cannot define this. A better term is 'well tolerated'.

RESPONSE: We have made this edit.

3. The authors should be clear in the abstract the second biopsy was taken after BAT alone and before nivolumab.

RESPONSE: We have added additional text to make this clear to the reader.

4. The authors should also make clear in the abstract there was a lack of correlation between BAT response and DNA repair defects, which they previously suggested.

RESPONSE: We have added this text.

5. The authors report that BAT induced changes in inflammatory pathways with the top pathway being the IL6-JAK-STAT3 pathway. It has been reported that IL6-JAK-STAT3 also has immunosuppressive functions on T cell and possibly drive cancer progression and treatment resistance. They should comment on this in their discussion. It would be important to show now BAT treatment impacted: 1) T cell exhaustion signatures in the RNAseq data; 2) Other inferred immune cell types; 3) Broader immune transcript expression using 770 gene Nanostring White paper.

RESPONSE: We performed a revised RNAseq analysis where we have *paired* the two samples (pre-treatment and on-treatment) for each patient and accounted for this in the analysis. You will now see a revised Figure 4 showing GSEA for *all* HALLMARK gene sets, rather than just those related to inflammation. We feel that this provides a better overall picture of changes in gene expression patterns. The top enriched pathway is now ALLOGRAFT REJECTION rather than the IL-6-JAK-STAT pathway. Thus, we did not discuss the IL6-JAK/STAT pathway. We have also included T cell exhaustion signatures and the signatures from the referenced Nanostring white paper. These are provided in the supplementary data. We do not make claims about tumor-infiltrating immune cell types inferred from the RNAseq in the manuscript. However, we performed cybersort on the data to address the reviewer's question. We present this data below for the reviewer, which shows that the most abundant cells inferred were CD4+ T cells. We think that direct measurement of abundance of infiltrating T cells by IHC, which is included in the paper, is more accurate than this method and therefore have omitted this cybersort analysis from the manuscript.

6. Authors report on the BAT induced inflammatory response; can they please comment on impact on T cell density (Figure 5). What does this mean for T cell anti-tumour function and the impact of BAT on nivolumab treatment outcome?

RESPONSE: In the discussion, we speculate that the pro-inflammatory gene expression changes induced by BAT may work in concert with the pre-existing increase in TILs in responders to facilitate a clinical response. Given that the T cells are predominantly PD-1+ T cells, it may be that nivolumab can help activate these T cells to further improve clinical outcomes. We have added some additional text to better state this.

7. One of 4 responder to nivolumab appears to have a mutation in an MMR gene. Did mutational burden/MMR mutational signature result in responders?

RESPONSE: We did not observe a correlation between objective or PSA response and pre-treatment TMB. Responder: TMB Median (Q1,Q3) = 1.9 (1.5,2.8) vs. Non Responder: TMB Median (Q1,Q3) = 1.3 (1.3, 2.1) P= 0.231. We did not measure mutational burden after treatment to determine if treatment resulted in a higher TMB in responders.

Minor comment

Typo on page 62 of the PDF: Cell Phenotypes "Evaluated" in T cell Panel

4.

RESPONSE: Thank you, we have corrected the typo.

Does PDL1 express in tumour, especially in responders please?

RESPONSE: We have not assess PDL-1 levels within the tumor.

[Editorial Note: prior referee report redacted]

Reviewer #5 (Remarks to the Author):

1) I think it is difficult to assess if the sequence BAT plus nivolumab has some clinical benefit as measured by response, given that most responses were observed in the BAT period alone. I think the trial should aim to show whether adding nivolumab extends time to progression or make those responses more durable, so maybe a randomised phase II of BAT vs BAT+Nivo trial and PFS endpoint would have reflected this more clearly. In these lines, maybe you want to add description of duration of responses (it is, as a minimum, defined for RECIST 1.1 in the guidelines, and you could also define for PSA). The swimmers plot helps hugely to understand the timing and duration of responses. It is also worth noting that 5 patients continue on treatment beyond radiological progression for a few weeks more – this is not uncommon in this disease (and particularly if patients are heavily pre-treated and don't have many options after). This is also highlighted in the PCWG3 guidelines, where the metric “no longer clinically benefiting (NLCB)” was introduced, defined as the date and the specific reason(s) a therapy was ultimately discontinued. It is somewhat subjective, but I think it would be worth reporting median time on treatment, as it may be more informative as a measure of clinical benefit. Please also add some data re compliance (dose reductions, whether patient could continue on one treatment only, did it happen?).

RESPONSE: Thank you for your feedback. We have now included the median time on study treatment to better inform the readers. The median time on study treatment was 5.6 months (95% CI: 5.6-8.4 months). No dose reductions of either drug were allowed. Patients were not able to stop the BAT and continue nivo or vice versa. N=2 patient experienced cardiac specific AE likely related to Nivo. These patients were removed from study as indicated and did not continue on BAT alone.

2) The trial has a 2-stage Simon design, but some details are missing (e.g. stage 1 and stage 2 success criteria, specifying this is a Minimax design). The statistical methodology used for the primary endpoint (95% exact confidence intervals used and p-value as a one-sample binomial test) does not take into account the 2-stage design – please note the following reference: Koyama T and Chen H (2008): Proper inference from Simon's two-stage designs. *Statistics in Medicine*, 27(16):3145-3154.

Appropriate p-value and confidence intervals could be calculated with the OneArmPhaseTwoStudy R package or clinfun R-package. For your trial, I have done these calculations using these packages, and it won't make a huge difference (p-value of 0.018), because the number of responses observed is well above the threshold of success for the design. But I wanted to make you aware as, if you had observed number of responses around the threshold of success, you could see some inconsistencies between the one-sample binomial test p-value and the two-stage p-value (like one-sample binomial p-value not significant when min number of responses was achieved). A similar issue occurs for the

confidence interval. These calculations also take into account if over-recruitment occurs, as it has been the case in the COMBAT trial.

RESPONSE: Thank you for your thoughtful suggestion and providing us the package information. We have re-analyzed the response rate using both packages that you were referring to: the OneArmPhaseTwoStudy R package and the clinfun R-package. We used the “twostage.inference” function within the clinfun R-package, with the code `twostage.inference(x=18, r1=5, n1=23, n=45, pu=0.25, alpha=0.1` (0.05 to calculate 95% CI)). The analysis yielded a p-value of 0.0189, with a point estimate of 40.2%, and a 95% CI: 27.7% - 51.1%. We agree that there were unimportant differences between these adjusted results and our original report of p-value of 0.02 (rounded from 0.0191), with a point estimate of ORR of 40.0%, and a 95% CI: 25.7% - 55.7%. In order to keep our results comparable with the reports of other studies that used Simon’s two-stage design but reported the naïve estimate, we would prefer to retain the original results. We would be happy to make the change if the reviewer insists. It’s worth noting that the OneArmPhaseTwoStudy package has been removed from the CRAN repository (<https://cran.r-project.org/web/packages/OneArmPhaseTwoStudy/index.html>).

3) Did some of these patients present with bone disease (in addition to soft tissue)? Should this be indicated in the baseline table? rPFS is defined as time to PD as per RECIST 1.1 in the manuscript, but the PCWG3 guidelines include progression on bone scan also as radiographic PD. Could you clarify if this is the case? Of note, the protocol specifies also as an endpoint PFS, a combination of clinical and radiographic progression (including progression on bone scan), which is not reported.

RESPONSE: We did not collect data on the number of patients with bone mets so this was not reported. Radiographic progression was determined per RECIST1.1 for soft tissue lesions and PCWG3 guidelines for bone lesions. We have updated our methods section to reflect this. We have now reported clinical or radiographic progression-free survival (crPFS). The median crPFS was 5.6 (95% CI: 4.8-6.0) months.

4) For these gene-expression analyses of the paired tissue samples, patients were stratified by clinical response to BAT and nivolumab (as either PSA50 or OR by RECIST). But the second biopsy is before nivolumab, so it would be worth clarifying/stating if these 6/12 responders did respond while on BAT only, or after when already on BAT+nivo, and maybe discuss results taking this into account. Would it be more appropriate to say then that “patients were stratified by their response to BAT”, only?

RESPONSE: Thank you for this comment. All 6 patients achieved a clinical response while on BAT alone. We have now made note of this in the results. However, 2 of the 6 had further PSA declines while on the nivolumab. We feel that the additional text added in the results section provides additional insights into this analysis for the reader.

5) In the comparison of COMBAT PSA response rates with TRANSFORMER, it may be helpful to report the PSA50 rate while on BAT only (16/45=35.6%), with its corresponding confidence intervals, which would more clearly overlap with those in the TRANSFORMER

study. As discussed above, in addition to rPFS may be worth to report median time on treatment and compare this to the TRANSFORMER study.

RESPONSE: Thank you – we have added the PSA50 RR for BAT alone along with the corresponding 95% CI (21.9%-51.2%). We have also added the median time on treatment, which was similar to the median rPFS - 5.6 months (95% CI: 5.6-8.4). We have added additional text about this when discussing the TRANSFORMER PFS.

Minor:

- Maybe plotting per-patient PSA dynamics (line or spaghetti plots) may also help visualise differences of PSA kinetics while in BAT only or while in the combination. Re Figure 1, it may help to have the same patients aligned, so the reader can map out per-patient max tumour shrinkage and max PSA decrease. The swimmer plot summarises nicely the time of the first decrease to qualify as a response, but not the magnitude of it. You could chose the order of the patients in the current PSA waterfall plot for the tumour shrinkage plot.

RESPONSE: We have added a figure in the supplement showing the PSA change and change in tumor sum matched with each patient to better demonstrate the relationship between PSA and objective responses. We have also added the median PSA PFS which was 4.0 months (95% CI: 3.0-5.0 months)

- Methods line 211 “The associations of patient baseline characteristics with rPFS and OS were evaluated using univariate Cox regression model.” could be changed to “The associations of EACH patient baseline characteristic with rPFS and OS were evaluated using univariate Cox regression models”, so it is clear these are separate Cox models for each factor.

RESPONSE: Thank you. We made this edit.

- Methods line 216: “All tests were two-sided unless otherwise noted, and statistical significance was set at $P < 0.05$ ” – this is except for the primary endpoint, as per design, statistical significance is one-sided $P < 0.1$.

RESPONSE: We’ve edited this sentence to: “All tests were two-sided unless otherwise noted, and statistical significance was set at $P < 0.05$, except for the primary endpoint, as per design, statistical significance was one-sided $P < 0.1$.”

- Results line 306: it says 8 patients experienced PD prior to start of nivolumab, but in the reasons for removal from study, for one patient reason is “Withdrew consent”. Assume it is withdrew consent AND progression? Was it a clinical progression without PSA and radiological progression?

RESPONSE: The patient did have a rise in $PSA > 25\%$ from baseline. This value was not confirmed on repeat as the patient withdrew consent. We provided additional details in the text to clarify.

- Results line 314: “One patient with a confirmed PSA50 response had “progressive disease” as best objective response due to the presence a new metastatic lesion.” Worth explaining timing of PSA respect timing of radiological progression, where these observed at the same time? Were target/non-target lesions responding/stable?

RESPONSE: We provided additional explanation in the text. This patient had a 50% decrease in the sum of target lesions concurrent with the PSA response. However, the new lesion resulted in this patient having progressive disease.

- Results line 318: a CR was observed, did patient had bone disease at baseline?

RESPONSE: We did not collect data on site of metastases except for the presence of

visceral mets. However, this patient did not have evidence of disease on a bone scan.

- Results line 321: I think it would be best to state upfront that, "Most responses were observed while on BAT monotherapy, with only N=2 confirmed PSA50 responses and N=3 objective responses occurred following the addition of nivolumab. (Figure 2)".

RESPONSE: We have made this edit. Thank you.

- Results line 322: Consider adding "further" - "Ten patients had a FURTHER decrease in PSA following nivolumab treatment (Extended Data Figure 1A, C)."

RESPONSE: We left this text as is. See next comment.

- Results line 324: "We observed N=4 patients that had a decline in PSA on nivolumab after experiencing a PSA rise on BAT". In Extended Data Figure 1, I only count 2 such patients (20&33), is this correct?

RESPONSE: Patients 9 and 23 had a decline in PSA after the addition of nivolumab. These declines in PSA did not go below the C1D1 value of PSA which occurred in patients 20 and 33.

- Results line 333: "11.1% (N=5/45) of patients remained on study for 12 months or longer." What do you mean, "on study"? "On study treatment"? I imagine patients are still on study after treatment discontinuation for OS. Also, note I count 4 patients on treatment for 12 months or longer in the swimmers plot, not 5.

RESPONSE: The 5th patient had radiographic progression just shy of the 12 months mark and was officially removed from study shortly after the 12 month mark. For the sake of transparency, we changed 12 months to 11 months.

REVIEWERS' COMMENTS

Reviewer #1 (Remarks to the Author):

The authors have addressed my comments.

I have no other comments to make.

Reviewer #5 (Remarks to the Author):

Minor comments:

- I would call crPFS just PFs, as in the protocol.

A point-by-point response to each reviewer's comments is listed below.

Reviewer #1 (Remarks to the Author):

The authors have addressed my comments. I have no other comments to make.

Reviewer #5 (Remarks to the Author):

Minor comments:

- I would call crPFS just PFS, as in the protocol.

RESPONSE: We have made this change.